# Endostatin and Cancer Therapy: A Novel Potential Alternative to Anti-VEGF Monoclonal Antibodies

**DOI:** 10.3390/biomedicines11030718

**Published:** 2023-02-27

**Authors:** Gabriel Méndez-Valdés, Francisca Gómez-Hevia, José Lillo-Moya, Tommy González-Fernández, Joaquin Abelli, Antonia Cereceda-Cornejo, Maria Chiara Bragato, Luciano Saso, Ramón Rodrigo

**Affiliations:** 1Molecular and Clinical Pharmacology Program, Institute of Biomedical Sciences, Faculty of Medicine, University of Chile, Santiago 8380000, Chile; 2Department of Biomedical Sciences, Humanitas University, 20090 Milan, Italy; 3Department of Physiology and Pharmacology “Vittorio Erspamer”, Faculty of Pharmacy and Medicine, Sapienza University, P.le Aldo Moro 5, 00185 Rome, Italy

**Keywords:** endostatin, angiogenesis, anti-cancer peptides

## Abstract

Angiogenesis is a physiological process that consists of the formation of new blood vessels from preexisting ones. Angiogenesis helps in growth, development, and wound healing through the formation of granulation tissue. However, this physiological process has also been linked to tumor growth and metastasis formation. Indeed, angiogenesis has to be considered as a fundamental step to the evolution of benign tumors into malignant neoplasms. The main mediator of angiogenesis is vascular endothelial growth factor (VEGF), which is overexpressed in certain cancers. Thus, there are anti-VEGF monoclonal antibodies, such as bevacizumab, used as anti-cancer therapies. However, bevacizumab has shown adverse events, such as hypertension and proteinuria, which in the most severe cases can lead to cessation of therapy, thus contributing to worsening patients’ prognosis. On the other hand, endostatin is an endogenous protein that strongly inhibits VEGF expression and angiogenesis and shows a better safety profile. Moreover, endostatin has already given promising results on small scale clinical studies. Hence, in this review, we present data supporting the use of endostatin as a replacement for anti-VEGF monoclonal antibodies.

## 1. Introduction

Cancer is one of the leading causes of death worldwide, being responsible for almost 10 million deaths in 2020, according to the World Health Organization (WHO), and with about 18 million new cases diagnosed every year [1]. This is one of the major public health problems in most countries around the world, being both a burden on health and economics.

Surgery can offer a total cure if the cancer is diagnosed at an early stage, however, regrettably most cancers are recognized when they are already locally advanced or with distant metastases, and in these cases, surgery alone does not provide an adequate treatment. In this scenario, the first line of treatment is chemotherapy. Chemotherapy can also be associated with radiotherapy [2], being more effective in improving the survival of cancer patients. Woefully, this therapy is accompanied by a wide variety of adverse effects that often result in discontinuation of it [2].

One of the fundamental hallmarks of cancer pathophysiology is angiogenesis, a physiological process that consists of the formation of new blood vessels from preexisting ones. Therefore, different strategies targeting this process have now been proposed and are currently used in clinical practice.

The therapy that is most studied and employed at the moment consists in monoclonal antibodies targeting the vascular endothelial growth factor (VEGF). These anti-VEGF monoclonal antibodies have been used as an adjuvant to standard therapies in many different cancer types, leading to an improvement of patients’ prognosis. One of the main downsides of this drug family is their toxicity. This issue in the safety profile of anti-VEGF therapy may lead to suspension of therapy. In order to overcome this problem, different options have been considered as possible alternatives, among them, endostatin, an endogenous anti-cancer peptide, has been singled out. In this review, the use of endostatin as a substitute for anti-VEGF monoclonal antibodies will be deepened.

## 2. Angiogenesis

Angiogenesis is a biological process that consists of the growth of novel capillary vessels from preexisting vasculature, providing tissues with oxygen and nutrients [3]. Capillaries are needed in all tissues for diffusion exchange of nutrients and metabolites. Changes in metabolic activity lead to proportional changes in angiogenesis and, hence, proportional changes in capillarity. Oxygen plays a pivotal role in this regulation.

There are two types of angiogenesis, sprouting angiogenesis and intussusceptive angiogenesis.

Intussusceptive angiogenesis involves formation of blood vessels by a splitting process in which elements of interstitial tissues invade existing vessels, forming transvascular tissue pillars that expand [4].

Sprouting angiogenesis is characterized by sprouts composed of endothelial cells, which usually grow toward an angiogenic stimulus such as VEGF-A. Sprouting angiogenesis can therefore add blood vessels to portions of tissues previously devoid of blood vessels.

This latter type of angiogenesis is one of the hallmarks of cancer. In this scenario, pathological angiogenesis is driven by the overexpression of pro-angiogenic factors, leading to an imbalance with antiangiogenic factors and the recruitment of a new vascular supply [5]. This process is triggered mainly by tissue hypoxia, which leads to expression of multiple growth factors via the hypoxia-induced factors (HIFs) by cancer cells and stromal cells recruited to the tumor [6]. This pathological process leads to the production of tortuous, frail, highly permeable blood vessels, and heterogeneous vascular zones on tumors [7,8].

One of the most important angiogenic inductors is vascular endothelial growth factor (VEGF). VEGF is a homodimeric glycoprotein with a molecular weight of approximately 45 kDa [9]. The family of growth factors and tyrosine kinase receptors include VEGF-A, VEGF-B, VEGF-C, VEGF-D, and placental growth factor (P1GF), of which VEGF-A is the major mediator of tumor angiogenesis [10]. It induces angiogenesis via a direct effect on endothelial cells, binding to two homologous tyrosine kinase receptors, VEGF receptor-1 (VEGF-R1) and VEGF receptor-2 (VEGF-R2), which are predominantly expressed on endothelial cells [9], but it can also be found on non-endothelial cells [11]. Despite the fact that VEGF-R2 is primarily involved in tumoral pathological processes such as tumoral angiogenesis, the activation of VEGF-R1 also plays an important role, including numerous mechanisms. The crucial ones are chemotaxis of inflammatory cells, secretion of inflammatory cytokines, the recruitment of medullary progenitor cells at the injury site, secretion of growth factors, interaction with PlGF, and activation of proteolytic enzymes [11].

Vascular endothelial growth factor gene expression is up-regulated by a number of factors, such as PDGF, fibroblast growth factor (FGF), epidermal growth factor (EGF), tumor necrosis factor (TNF), etc. [9]. VEGF is also up-regulated by HIF [12]. Vascular endothelial growth factor’s signaling through VEGFR1/R2 regulates the activities of several kinases, leading to cell proliferation, migration, survival, and vascular permeability during angiogenesis [12]. It plays an important role in pathological angiogenesis, inducing the development and progression of certain pathological conditions, including tumor growth and metastasis [11]. Solid tumors need an appropriate blood supply to grow, without it, they can reach to approximately no more than 106 cells, due to lack of oxygen and nutrients. Tumors in their avascular phase can remain latent, maintaining a state between cell proliferation and apoptosis. VEGF acquires a key role when there is the tumor conversion to an angiogenic type, allowing its access to oxygen and nutrients and growth [8]. VEGF is expressed in most types of cancers, and its increased expression is usually associated with a less favorable prognosis [10].

## 3. Anti VEGF Monoclonal Antibodies

The angiogenic pathway is composed of different steps, consequently each step can be taken as a target to create a novel anti-humoral therapy. Antiangiogenic agents targeting the VEGF pathway include monoclonal antibodies against VEGF, tyrosine kinase inhibitors, VEGF decoy receptor, and specific inhibitors against VEGFR-2 [13]. Among the different classes of drugs, the most widely used are monoclonal antibodies. A well-known anti-angiogenic monoclonal antibody is bevacizumab [13].

Bevacizumab is a humanized monoclonal antibody that binds to circulating VEGF-A isoforms and thereby inhibits their binding to cell surface receptors. This results in an inhibition of the activation of the VEGF pathway, causing a decrease in tumor blood flow and blood vessel growth [14]. Thanks to its mechanism of action, bevacizumab leads to a reduced angiogenesis and growth of several tumors, both primary and metastatic. This monoclonal antibody is intravenously administered and can be used for the treatment of colon, lung, glioblastoma, and renal carcinomas [15].

There are several studies showing the advantage of using bevacizumab as an adjuvant therapy to chemotherapy, in particular in colorectal cancer and lung cancer. In a study conducted by Goldberg et al. [16] for the first-line treatment of metastatic colorectal cancer, the addition of bevacizumab to bolus irinotecan, leucovorin and fluorouracil (IFL) chemotherapy conferred a clinically meaningful and statistically significant benefit for all study endpoints, including overall survival, progression free survival, and response rate, and was associated with an acceptable side effect profile. Furthermore, Ferrara et al. [17] conducted a study where patients with previously untreated advanced non-squamous and non-small cell lung cancer received bevacizumab in combination with standard chemotherapy (paclitaxel and carboplatin). Preliminary results from this large, randomized Phase III clinical trial showed that the patients treated with bevacizumab as adjuvant therapy lived longer than patients who received the same chemotherapy without bevacizumab.

Anti-VEGF agents present high variability in their clinical response. Recent studies reported that patients with metastatic colorectal cancer who were also carriers of the VEGF-A rs699947 A/A allele, a single nucleotide polymorphism, had markedly higher PFS and OS after bevacizumab administration. Moreover, carriers of ICAM rs1799969 G/G and BRAF rs113488022 mutant (78,3%, 26%, and 8.6% of the studied population, respectively) presented a marked increase in their OS [18].

Another study in metastatic colorectal cancer reported an association between the allele CXCR1 rs2234671 G>C and a lower relative risk of metastatic cancer. A correlation between different alleles and the outcome based on race has also been made. In particular, the alleles CXCR2 rs2230054 T>C and EGF rs444903 A>G (71%, 38%, and 44% of the patients were homozygous, respectively) were associated with a better clinical outcome and tumor response in Caucasian patients, but not Asian and Hispanics [19].

In advanced breast cancer patients, a considerable improvement of median OS was associated with VEGF-2578/-1154 AA/AA alleles, present on 7.6% of patients, and most common alleles showed worse prognosis [20]. In addition, Etienne-Grimaldi et al. [21] reported that higher TTP was associated with VEGF-936C>T with the 936T, carried by 23.3% of the patients analyzed, and the VEGF-A -634 G > C polymorphism was associated with a higher toxicity score, with -634C allele carriers likely to develop hypertension and thromboembolic events. Exposure to this monoclonal antibody has been correlated to OS in patients with metastatic colorectal cancer, where higher doses showed better outcomes [22], which could be related to the complex pharmacokinetic behavior of this drug.

### Adverse Events

Anti-VEGF agents, in particular bevacizumab, were introduced in oncology to inhibit tumor-induced angiogenesis feeding neoplastic tissues. These agents are administered both systemically and intravitreally.

When agents are administered intravitreally locally into the eye globe, they are injected in much smaller quantities [23]. Even if the quantity of the drug administered is lower, side effects are still present. This happens because the presence of side effects is neither due to the route of administration or to the quantity of the drug given, but they are consequences of the suppression of the pathways that regulate the maintenance of the microvasculature and that inhibit anti-VEGF therapy [24]. Consequently, systemic exposure has been seen with intravitreal administration even at low doses [25]. The incidence of these adverse effects vary according to the drug used and range from those with high frequency, such as hypertension and proteinuria, to those with lower frequency, such as impaired wound healing, gastrointestinal perforation, hemorrhage and thrombosis, reversible posterior leukoencephalopathy, cardiac impairment, and endocrine dysfunction [24].

In relation to intravitreal anti-VEGF therapy used in retinal disorders specifically, the systemic effects are the same as those previously mentioned with the addition of ocular complications, the incidence of which again depends on the drug used. Examples of ocular complications include endophthalmitis, intraocular inflammation, rhegmatogenous retinal detachment, intraocular pressure elevation [26].

As previously stated, angiogenesis is carried out by the signaling of a family of five members, VEGF-A, VEGF-B, VEGF-C, VEGF-D, and P1GF, which are necessary to carry out processes such as wound healing, vascular permeability, and embryonic vasculogenesis. All these processes can be exploited by the tumor to guarantee its growth in situ and its ability to generate metastasis.

Thus, blocking the VEGF signaling pathway through various drugs, such as monoclonals (mAbs), ligand inhibitors, and TKIs inhibitors, have turned out to be beneficial in various types of cancers [27].

However, by inhibiting VEGF signaling pathways, physiological cell functions cannot be properly carried on, which is why the emergence of adverse effects after administration of such therapies is seen. Thus, several studies have shown that common adverse events such as hypertension are associated with decreased production of nitric oxide (NO) in arterioles and other vessels. Moreover, proteinuria can be explained through the blockade of VEGFR-2 signaling in glomerular capillaries, decreasing the amount of fenestrations [24].

In the angiogenesis of an adult organism, VEGFR-1 does not play a relevant role in physiological functions, but it has an important role in angiogenesis related to tumor growth and nutrition.

On the other hand, VEGFR-2 fulfills pro-angiogenic functions in healthy and damaged adult tissues and it is also involved in tumorigenesis.

Currently, approved therapies against various solid tumors hamper VEGF-A (bevacizumab, aflibercept), VEGFR-2 (ramucirumab) signaling or VEGF receptor activation.

Indeed, the problem with the emergence of adverse events resides in the fact that interfering with VEGF-A/VEGFR-2 signaling also inhibits normal physiological processes carried out by the activation of this receptor [28]. Clinical studies have shown that anti-VEGF monoclonal antibodies have an associated toxicity, in addition to the toxicity that chemotherapy already has.

In a phase 3 RCT, 903 patients with renal cell carcinoma were divided into two groups: 451 patients received standard therapy plus sorafenib, a VEGFR inhibitor, and 452 standard therapy with placebo. Although the PFS showed a statistically significant benefit of the first group over placebo, there was a significant difference between the sorafenib group and the placebo group in the frequency of serious adverse events (34% and 24%, respectively; *p* < 0.01). In the sorafenib group, 76 patients had hypertension of any grade, in comparison with the placebo group who were only eight. Sorafenib was also associated with other adverse events, such as gastrointestinal and dermatologic [29].

In another phase 3 RCT, Wakelee et al. administered chemotherapy (arm A) and chemotherapy plus bevacizumab (arm B) in patients with resected non-small cell lung cancer. In this study, 8.4% of patients from arm A and 27.6% of patients from arm B had to discontinue therapy because of adverse events. Additionally, hypertension was more common in patients treated with bevacizumab [30].

Given what was previously exposed, even if anti-VEGF monoclonal antibodies seem to improve survival, they still have an important toxicity that can manifest in different grades in different patients.

These findings make the investigation of new drugs with a better safety profile and equal or better therapeutic outcomes crucial. This is why in this review we evaluate the use of endostatin, a peptide also related to angiogenesis, as a cancer treatment.

## 4. Endostatin

Endostatin constitutes one of the most studied peptides with inhibitory effect on angiogenesis [3]. This peptide is a 20 kDa C-terminal cleavage fragment from the α1 chain of type XVIII collagen, which is an extracellular matrix protein recognized for its anti-atherosclerotic effect as well as a potent inhibitor of angiogenesis [3]. Endostatin has been artificially synthesized in a recombinant human form with the addition of nine extra amino acids that confer greater stability, solubility, and antiangiogenic effect [31]. The endostatin mechanism of action is not completely understood. Evidence shows that this molecule exerts its angiostatic effect through multiple mechanisms involving elements of the extracellular matrix, as well as proteins and signaling cascades related to endothelial cell migration and proliferation.

First, it has been recognized that endostatin is capable of inhibiting matrix metalloproteinases (MMPs), particularly MMP-2, MMP-9, and MMP-13, which due to their proteolytic action, facilitate the migration and invasion of endothelial cells during the process of angiogenesis.

Additionally, endostatin through α5β1-integrin binding inhibits the FAK/Ras/p38-MAPK/ERK signaling cascade, with suppression of HIF-1α/VEGF-A, and thus leading to an inhibition of endothelial cell migration. Endostatin can also induce autophagy in endothelial cells [32], through a signaling pathway mediated by activation of the Src family of kinases [33].

Moreover, endostatin is responsible for the down-regulation of β-catenin dependent on Wnt signaling. This latter mechanism is related to a suppression of the transcription of important genes involved in the cell cycle, which is strongly linked to endothelial cell apoptosis.

Alternatively, endostatin binds directly to VEGF-R2, blocking the action of VEGF, thereby suppressing vascular endothelial tube formation [34]. These mechanisms are summarized in Figure 1. In consistency with the angiostatic profile of endostatin, in vitro models using human umbilical endothelial vein cells have proved that this agent has the capacity to inhibit the migration of endothelial cells, as well as the formation of capillaries and the interconnection amongst them [35]. As a consequence of these findings, and considering angiogenesis as one of the hallmarks of cancer, this agent has been tested in several experiments in order to examine whether it is able to decrease tumor growth and prevent metastasis.

Preclinical findings in a xenograft murine model of lung cancer demonstrated that endostatin is able to normalize the structure and function of the tumor vasculature, characterized by a decrease in microvessel density and improved vessel wall structuring [36]. In the same study, it was demonstrated that this effect was translated into an improved delivery of cytotoxic agents to the tumor area, therefore endostatin would have a synergistic effect when administered simultaneously with cytotoxic drugs [36]. Moreover, it has been shown that endostatin administered alone is able to decrease the number and size of metastatic pulmonary nodules, which correlates with prolonged overall survival in mice treated with this agent [37]. It has also been tested in models of ovarian cancer, whose characteristics are its aggressiveness and high metastatic capacity. The oncological patients who received the treatment had a marked reduction in tumor growth, coupled with a decrease in the migratory capacity of the tumor cells. In addition, this was correlated both with a lower expression of angiogenesis-promoting proteins and with a lower epithelial–mesenchymal transition [38].

One of the major problems in the use of endostatin is that the dose administered to make this compound biologically active is very high. Thus, it has to be produced in great quantities, which makes it also a possible economic burden. Another problem contributing to this issue is that its purification process could denature its structure and lower its yield. Indeed, the endostatin production system could disable the appropriate folding in the production of the soluble protein, affecting its bioactivity. Lastly, this peptide has a short half-life [39], which leads to the necessity of being continuously infused over long periods.

Most of the studies of endostatin in humans have been carried out in patients with locally advanced or metastatic cancer, where endostatin was added to the first line of treatment, usually chemotherapy or chemoradiotherapy.

The following sections will explore the results of the most relevant studies done in humans by using endostatin as a treatment for different types of cancer, emphasizing its safety profile.

### 4.1. Pulmonary Cancer

Most studies using endostatin have targeted patients with non-small cell lung cancer (NSCLC), which accounts for 85% of all lung cancers. A large portion of this type of cancer is diagnosed in advanced stages (III or IV), where the first line of treatment is platinum-based chemotherapy together with new generation cytotoxic agents, such as etoposide and gemcitabine and/or radiotherapy [40].

Multiple clinical trials have been conducted in recent years demonstrating that the addition of endostatin to the standard treatment of advanced NSCLC significantly increases the overall survival (OS) and the progression-free survival time (PFS) of patients in locally advanced [41,42,43,44,45,46,47,48,49,50,51] or metastatic disease [43,49,50,51,52,53,54]. Trials have also been tested on small cell lung cancer, giving promising results in survival and tolerability of the therapy [55,56,57].

Interestingly enough, the addition of endostatin to these different trials did not increase the incidence of adverse events in a significant manner in comparison to their respective control groups. Among the adverse effects that have been observed in combined treatment schemes of endostatin, in addition to chemotherapy or radiochemotherapy, there are hematological toxicity (neutropenia, thrombocytopenia, and anemia), nausea, vomiting, and fatigue [57]. Fortunately, none of these adverse effects led to discontinuation of therapy.

In addition, other authors have compared the efficacy and safety of using extended endostatin regimes during concomitant chemotherapy [43,51]. The results indicate that the extended endostatin treatment not only is associated with a significant improvement in patient prognosis, but also that it does not lead to a marked increase in the most common adverse effects associated to the therapy. The only exception is represented by cardiac disorders and hypertension, which are increased in patients who received an extended endostatin course, although none of these resulted in discontinuation of therapy [43,51].

### 4.2. Gastric Cancer

Gastric cancer accounts for more than 1 million cases diagnosed each year worldwide. Those diagnosed in stage IA or IB have a 5-year survival of between 60–80%. Unfortunately, most of them at the time of diagnosis are already in the metastatic stage, which drastically worsens the prognosis, with a 5-year survival as low as 18% for those diagnosed at stage III [58]. In this context, the first line of treatment is chemotherapy with the FOLFOX (folic acid with 5-Fluorouracil and oxaliplatin) or CAPOX (capecitabine and oxaliplatin) regimens with or without trastuzumab, in cases where HER2 is overexpressed [58]. Recently, clinical guidelines recommend that ramucirumab, a VEGF-R2 monoclonal antibody, can be added to the treatment regimen in the context of disease progression [59]. There are few human studies that have used endostatin for the treatment of gastric cancer. These studies mostly involve patients with locally advanced disease or with distant metastasis.

In a study by Yao J et al. [31] involving 33 patients with advanced gastric cancer with peritoneal carcinomatosis, it was shown that treatment with endostatin plus chemotherapy was superior to chemotherapy alone, with a significantly higher median OS in the group that received endostatin compared to the control group (15.8 vs. 9.8 months).

The main side effects that led to dose decrease and discontinuation of therapy were neutropenia and severe thrombocytopenia. However, they were present in both groups with no significant difference. The adverse events associated with endostatin only correspond to hypertension and bleeding, but they did not translate into discontinuation of therapy.

In another clinical trial conducted by Yang H et al. [60], endostatin plus SOX (S-1 with oxaliplatin) was shown to be effective in the treatment of liver metastases in patients with gastric cancer. The most common adverse effects, such as gastrointestinal reaction, hematological toxicity, and cardiac disorders, were equally present in both groups, but it is interesting to highlight the fact that the severity of these adverse reactions were lower in the group that received endostatin.

One of the most important clinical trials studied the safety and efficacy of molecular targeted therapy in patients with advanced gastric cancer [61]. A total of 200 patients were divided equally into four groups, comparing a control group receiving chemotherapy with others receiving bevacizumab (VEGF-R monoclonal antibody), apatinib (tyrosine kinase inhibitor), and endostatin. The results indicated that molecular therapies were significantly more efficient than chemotherapy in reducing and controlling tumor lesions, and that there were no significant differences between the three experimental groups.

The main adverse effects that occurred in all groups were neutropenia, nausea, vomiting, and rash, but their incidence was significantly higher in the control group. The results of this study lead to the conclusion that molecular targeted therapies represent the treatment of choice for patients with advanced gastric cancer, due to a higher efficacy in the control of tumor lesions and a lower frequency of therapy-related adverse effects.

### 4.3. Esophageal Cancer

Esophageal cancer accounts for about 5% of cancer deaths worldwide, with an estimated 570,000 cases diagnosed in 2018, with squamous cell esophageal cancer (SCEC) being the most frequent subtype [62]. It is a cancer with a poor prognosis, given that more than 70% is diagnosed at an advanced stage, with a 5-year survival rate of 25.1% when there is local-regional dissemination and merely 4.8% in the setting of distant metastases [62]. For patients with locally advanced disease, the first line of treatment is surgery accompanied by neoadjuvant chemoradiotherapy prior to surgery, however, for patients with metastases at the time of diagnosis, chemotherapy or chemoradiotherapy is the preferred treatment alternative [63].

The first clinical studies on endostatin in esophageal cancer showed that the endostatin plus DP (docetaxel and cisplatin) regimen did not present a higher rate of the most common adverse effects than the DP regimen alone.

It is noteworthy to point out that three of the ten patients in the group that received endostatin presented ECG changes (T wave changes). However, these changes normalized at the end of the treatment cycles [64].

In the phase II study conducted by Hu Z et al. [65] involving 50 patients who received a regimen of endostatin plus irinotecan/cisplatin, the efficacy and safety of this regimen for the treatment of advanced SCEC was demonstrated. The median PFS was 4.01 months, and the median OS was 12.32 months.

The most frequently reported side effects were leukopenia (18.0%) and neutropenia (16.0%); they presented with severity of grade 3 or higher in five patients (10.0%), leading to the discontinuation of treatment.

Similarly, another phase II clinical trial conducted by Wang Z et al. [66] demonstrated the safety and efficacy of endostatin plus paclitaxel/nedaplatin. The study was conducted on 53 patients with locally advanced or metastatic esophageal squamous cell cancer. The most frequently observed grade 3 or higher adverse effects were neutropenia (17.0%) and anemia (3.8%), and no treatment-related deaths occurred during the entire duration of the study.

Another study worth to be mentioned is the one made by Zhong Z et al. [67], where endostatin therapy plus chemoradiotherapy was shown to be superior to chemoradiotherapy for the treatment of locally advanced but not distant metastatic SCEC. The 1-year and 3-year overall survival rate was significantly higher in the endostatin group than in the radiochemotherapy alone group (72% vs. 50%, 32% vs. 22%, respectively), and the median PFS was 11.3 months for the endostatin group and 8.1 months for the control. Furthermore, there was no treatment-related toxicity that could be directly attributed to endostatin. Indeed, the most common adverse effects observed were probably associated with chemoradiotherapy.

### 4.4. Colorectal Cancer

Colorectal cancer is the third most common cancer worldwide, reporting 1,931,590 new cases in 2020 [68].

The most common treatments for metastatic colorectal cancer correspond to infusions of FOLFIRI (5-Fluorouracil with leucovorin and irinotecan) or FOLFOX (5-Fluorouracil with leucovorin and oxaliplatin) schemes, which have been combined in recent years with anti-VEGF or EGFR antibodies [69].

Bevacizumab is an example of an anti-VEGF antibody, which is approved for first-line use in colorectal cancer.

Although its effectiveness given as a monotherapy has been demonstrated, a meta-analysis reported that there is insufficient evidence to support its use as an adjuvant therapy with other regimens, such as FOLFIRI or FOLFOX.

In addition, its association increased the frequency of adverse events such as hypertension, proteinuria, bleeding, and thromboembolism, along with an increase in treatment interruption [70].

Various studies have tested the use of endostatin in combination with different common chemotherapeutic schemes against colorectal cancer, such as FOLFOX4 [71], modified FOLFOX6 [72], FOLFIRI [73], or including several of them [74].

Li et al. conducted a randomized controlled trial to evaluate the efficacy and safety of the use of endostatin plus FOLFIRI chemotherapy in patients with advanced colorectal cancer, reporting a significantly higher ORR (42.9 vs. 29.4%) and PFS (14.5 vs. 11 months) than the control group [73].

Moreover, Xu et al. evaluated the use of endostatin in combination with FOLFOX4 in patients with non-metastatic colorectal cancer through a retrospective controlled study, reporting a significantly higher ORR (38.9 vs. 22.3%), PFS (6.4 vs. 3.8 months), and OS (12.1 vs. 11.4 months) than the control group [71].

In addition, a pilot study evaluated the efficacy and safety of the use of endostatin in combination with different chemotherapy schemes (CAPIRI, GP, XELOX, DCF, FOLFIRI, or FOLFOX4) in patients with metastatic colorectal and gastric cancer, reporting an OS of 10.3 months (95% CI, 3.9–16.7 months), median time to progression of 2.6 months (95% CI, 2.0–3.2 months), disease control rate of 47.6%, and a ORR of 19.0%, and in patients treated with first-line therapy, the response rate was 57.1%.

Another interesting aspect is that endostatin was also given to a small cohort of patients (*n* = 5) in addition to previously failed third-line therapies. The results of the study showed disease stability with a maximum TTP of more than 11.0 months. Therefore, the authors indicate that the association of endostatin and chemotherapy could also be able to reverse chemo-resistance. This is an aspect about endostatin treatment that surely needs to be investigated [74].

Regarding the safety of endostatin in patients with colorectal cancer, a phase I clinical trial evaluated the safety, tolerability, and pharmacokinetics of the use of endostatin in combination with the modified chemotherapy regimen FOLFOX6 as a first-line treatment in patients with advanced colorectal cancer using a dose-escalation methodology.

Among the results, it was reported that the most frequent drug-related adverse events were leukopenia, neutropenia, anemia, anorexia, ST-segment/T wave changes, and nausea, but those that presented with a severity grade of 3–4 were only neutropenia, leukopenia, and thrombocytopenia.

There have also been two patients in the endostatin group that stopped therapy after an episode of ventricular arrhythmia [72].

Nonetheless, these results are similar to those reported in controlled clinical trials, where hematological and gastrointestinal adverse events are the most frequent, with no significant differences between the endostatin plus chemotherapy group and the control group [71,73].

Even so, cardiac adverse events vary between studies, where Zhou et al. reported that three patients presented transient sinus bradycardia with spontaneous remission [74]; Xu et al. that 17.7% presented hypertension and 11.1% cardiac ischemia, both in grade 1–2 vs. 5.6% and 0.0%, respectively, in the control group [71]; and Li et al. that three patients presented grade 1 electrocardiogram abnormalities, reverted by the administration of fructose diphosphate sodium, and two presented grade 1–2 hypertension, which was reversible and manageable [73].

### 4.5. Nasopharyngeal Cancer

Nasopharyngeal carcinoma (NPC) is a relatively uncommon cancer in comparison with the others. It had an estimated global incidence of 129,000 cases for the year 2018 [75]. NPC generally responds favorably to radiotherapy, with intensity-modulated radiotherapy (IMRT) the first line of treatment in stage I disease. Patients with locally advanced or metastatic disease (stage II-IV) benefit from adding chemotherapy to IMRT [75]. Existing studies using endostatin in this form of cancer are focused on comparing whether the addition of this agent to conventional therapy represents an improvement in long-term outcomes with an acceptable safety profile, given that the existing first line of treatment to date is very effective in improving short- and medium-term survival.

A study conducted by Guan Y et al. [76] involving 22 patients with stage III-IV NPC, evaluated the safety profile of the endostatin plus IMRT scheme associated with chemotherapy.

The results indicated that this regimen was not associated with a higher rate of adverse effects than those historically reported for standard treatment with IMRT plus chemotherapy. Moreover, endostatin treatment was associated with a lower incidence of nasopharyngeal mucosal necrosis/infection compared to literature reports for the treatment of stage III-IV NPC (31.8% vs. 40.6%, respectively).

A multicenter phase II clinical trial performed by Li Y et al. [77], involved 114 patients with stage III-IV NPC in order to determine the efficacy and safety of endostatin treatment. The experimental group received endostatin plus IMRT with chemotherapy, while the control group received IMRT plus chemotherapy only.

After an average follow-up of 67 months, the results indicated that the experimental group showed a slight but significant improvement in ORR at 3 months after treatment, however, this did not translate into significant differences in the curative effect on nasopharyngeal lesions at long-term follow-up. At 5 years follow-up, there were no significant differences in the OS, PFS, distant metastasis-free survival (DFMS), and locoregional failure-free survival rates between the two groups.

In this study, there was no toxicity associated with endostatin treatment, and the frequency of adverse effects also showed no significant differences.

On the contrary, a retrospective study demonstrated that treatment with endostatin associated with chemoradiotherapy was associated with a significant improvement in PFS and distant metastasis-free survival rates at 3-year follow-up compared to chemoradiotherapy alone (81.4% vs. 63.6% and 88.3% vs. 77.3%, respectively), although no improvement in OS was found [78].

Another retrospective study conducted by Chen W et al. [79] compared the efficacy and long-term adverse reactions between IMRT plus endostatin and IMRT plus chemotherapy. The results indicated that the IMRT plus endostatin group had no significant differences in long-term efficacy at 5-year follow-up in terms of OS, PFS, and DMFS ratios. Nonetheless, the IMRT plus endostatin group was notable for a substantially more favorable long-term adverse effect profile, with a significant decrease in both the incidence and severity of xerostomia, mouth-opening difficulty, and soft tissue fibrosis.

### 4.6. Breast Cancer

Currently, breast cancer is the most common cancer worldwide [80]. Although there are different histological types, triple negative breast cancer (estrogen receptor, progesterone receptor, and human epidermal growth factor receptor 2 are all negative) stands out for its aggressive biological behavior, low response to treatment, and poor prognosis [81,82]. However, in recent years, the efficacy of treatments and the survival advanced breast cancer patients of the different molecular subtype have improved, mainly due to the deepening of the use of personalized strategies based on anti-VEGF monoclonal antibodies, tyrosine kinase inhibitors, immune checkpoint inhibitors, CDK4/6 inhibitors [83].

The inclusion of anti-VEGF monoclonal antibodies to breast cancer therapy has shown promising results on ORR and PFS, but not on OS [84].

The use of endostatin has shown efficacy and safety when combined with chemotherapy in patients with triple negative breast cancer [85].

Various clinical studies have reported encouraging results when evaluating the efficacy and safety of the use of endostatin in combination with classical therapy in patients with different subtypes or clinical stages of breast cancer, obtaining high overall response rates and overall survival, without increasing the frequency of adverse events [84,86,87,88].

A prospective study evaluated the use of endostatin in combination with taxanes-based chemotherapy in patients with HER-2 negative metastatic breast cancer, reporting an overall response rate of 68.4%, having a greater response in those patients who received the therapy as first line (79.3%) versus those who received it as second and third line or beyond (54.5% and 16.7%, respectively). Additionally, those patients who had not been previously treated with taxanes showed a higher overall response rate. Regarding the PFS, the median was 10.8 months [86].

Moreover, another prospective study evaluated the use of endostatin in combination with platinum-based chemotherapy in patients with triple-negative breast cancer, reporting an ORR of 47.6%, a PFS of 8.8 months (95% CI: 7.2–10.4 months), and a median overall survival of 13.3 months (95% CI: 11.6–15.0 months) [87].

Another study worth mentioning is that of Chen et al. They carried out a phase 3 clinical trial to evaluate the use of endostatin with docetaxel and epirubicin as first-line therapy in patients with stage IIA-IIIC breast cancer [88]. In this study, the authors established the clinical and pathological response as the primary endpoint, defining an objective response as those patients who had a disappearance of all target lesions or at least a 30% decrease in the sum of the longest diameter of target lesions and a pathological response as those patients who had no residual viable invasive tumor. An objective response of 91.0% and pathological response of 10.7% was reported as results in patients who received endostatin plus chemotherapy vs. 77.9% and 7.7%, respectively, in those who received chemotherapy alone [88].

Additionally, there have been studies where endostatin effectiveness was studied as a monotherapy. In a phase II clinical study, the use of endostatin alone for patients with TNM stage III breast cancer was evaluated. Patients were randomized to neoadjuvant therapy consisting of endostatin in combination with docetaxel, epirubicin, and cyclophosphamide or chemotherapy alone.

As major findings, the authors reported significant differences in the ORR, with 81.82% for patients who received endostatin plus chemotherapy vs. 58.14% for those who received only chemotherapy, highlighting that those patients with infiltrating ductal carcinoma showed higher sensitivity to treatment with endostatin. In addition, the median OS was significantly higher in the endostatin group 74.2 months vs. 59.1 months, which was also reflected in the 3- and 5-year OS rates. Finally, the reported median relapse-free survival was 67.3 months vs. 55.0 months in the control group [84].

Finally, regarding the evidence from prospective studies on adverse events by associating endostatin with chemotherapy treatment, Huang et al. reported neutropenia (80.7%), leukopenia (77.2%), liver dysfunction (10.5%), and peripheral neurotoxicity (8.8%) as the most frequent adverse events in grade 3–4, while Tan et al. reported neutropenia (14.3%), anemia (14.3%), leukopenia (9.5%), thrombocytopenia (9.5%), febrile neutropenia (4.8%), and hypertension (4.8%) as grade 3–4 adverse events [86,87].

Furthermore, randomized controlled trials have reported no significant differences in the frequency of adverse events between groups receiving endostatin plus chemotherapy and those receiving chemotherapy alone [84,88]. A summary of these articles can be found in Table 1.

## 5. Discussion

Cancer remains one of the leading causes of death worldwide and a major public health challenge. In particular, what makes their management difficult is the fact that most cancers are diagnosed in advanced stages of the disease. In this scenario, chemotherapy takes a major role as the first line of treatment.

In the last few years, in the oncological field, there has been an increase in the employment of targeted therapies, thus the molecular markers and pathophysiological events of the tumor gained more importance than the tumor location.

As abundantly stated in this review, one of the most important processes for the development of a tumor is angiogenesis. Angiogenesis in cancer provides the blood supply for tumor growth and the ability to generate metastases. Indeed, its blockage has acquired prior importance as an object of studies focused on preventing tumor progression.

There are already several therapies that are added to the conventional chemotherapeutic scheme.

In particular, anti-VEGF therapies, aimed to block signaling pathways that promote angiogenesis, are widely used. Anti-VEGF therapies improve parameters, such as OS, PFS and ORR, but they also bring about various adverse effects, ranging from frequent events, such as hypertension and proteinuria, to less frequent ones, such as impaired wound healing, gastrointestinal perforation, hemorrhage and thrombosis, reversible posterior leukoencephalopathy, cardiac impairment, and endocrine dysfunction.

Given the above, the use of combined therapies brings about a dilemma concerning the balance between the therapeutic benefits and the patients’ quality of life.

Currently, there are novel drugs that seem to have the same or better efficacy than the classic chemotherapeutic/anti-VEGF, but very few studies have been made in order to understand if these drugs can permanently replace anti-VEGF therapies.

In a trial performed by Chen et al. [94], platinum-based doublet chemotherapy in combination with endostatin had similar OS and PFS, but less severe adverse reactions than the same regime in addition to bevacizumab in patients with advanced NSCLC. The limitations in this study may explain the lack of statistical significance, leaving the true effectiveness of this drug uncertain.

In a meta-analysis performed by Shi et al. [95] on the impact of angiogenesis inhibitors in survival of patients with SCLC, only an improvement of PFS by bevacizumab was found. As evidence and works in this area are scarce, only nine articles could be analyzed, including nonrandomized clinical trials, where there was a single study testing endostatin.

Most of the studies using endostatin for the treatment of different types of cancer focus on patients with advanced stage disease, mostly stage III-IV, in order to evaluate whether this agent is able to improve survival and also have an acceptable safety profile that does not result in treatment discontinuation.

According to the different RCTs performed in lung, gastric, esophageal, colorectal, and breast cancer, treatment with endostatin associated with conventional therapy represents a significant improvement in the prognosis of patients in comparison to the first line of treatment with chemoradiotherapy or chemoradiotherapy alone [41,44,48,49,54,67,71,73,88,89,91]. Moreover, the studies conducted by Hu W et al. and Zhao J et al. demonstrated that the use of extended endostatin schedules, meaning 4 or more cycles, is not associated with a higher rate of adverse effects and correlates with an improved prognosis [43,51]. The exception to this pattern is the case of nasopharyngeal carcinoma, in which endostatin treatment correlates only with a better ORR at 3 months, but this does not translate into an improvement in long-term survival parameters, according to a phase II clinical trial performed by Li Y et al. [77].

However, a retrospective analysis indicates that although endostatin treatment in nasopharyngeal carcinoma is not associated with a significant improvement in 3-year OS, it does represent an improvement in PFS and DMFS [78]. The authors suggest that this discrepancy could be due to the fact that first-line treatment consisting of chemoradiotherapy is highly effective, even for patients with advanced stage nasopharyngeal carcinoma [77,78].

Regarding the safety profile of endostatin, most studies indicate that it is not associated with an increase in the rate or severity of the most frequent adverse effects associated with conventional therapy, which are usually hematological toxicity and gastrointestinal reaction [31,57,60,61,65,76,84,88]. The most frequently reported grade 3–4 adverse effects were leukopenia, neutropenia, thrombocytopenia, nausea, and vomiting, but these were related to the treatment with radiation and/or cytotoxic agents and not to endostatin [61,67].

In the study performed by Yang H et al. [60] targeting metastatic gastric cancer, it was observed that the addition of endostatin to the conventional regimen was associated with a lower degree of severity of chemotherapy-related adverse effects. In addition, endostatin treatment in patients with nasopharyngeal carcinoma was associated with a lower incidence of nasopharyngeal mucosal necrosis/infection [76], but also with a marked reduction in the frequency and severity of long-term sequelae associated with radiotherapy, such as xerostomia, mouth-opening difficulty, and soft tissue fibrosis [79].

The main adverse effects that have been directly attributed to endostatin therapy are cardiovascular alterations. Different reports indicate that the most frequent would be hypertension, ECG abnormalities (ST/T wave changes), cardiac ischemia, and transient sinus bradycardia, which occur more frequently in hypertensive patients or those with a history of coronary heart disease [3]. However, none of them resulted in discontinuation of therapy, being easily manageable or with spontaneous remission after completing the treatment cycles [55,64,73]. An in vivo/in vitro study made by Guan et al. [96] showed that dihydromyricetin could protect against this adverse reaction. The trial conducted by Chen et al. [72] on colorectal cancer was the only study in which two patients in the group receiving endostatin had to discontinue treatment after the occurrence of an episode of ventricular arrhythmia.

In conclusion, the current problems with studies concerning endostatin seem to be the lack of them and, among those that currently exist, their low number of patients, decreasing their statistical power and the absence of these in other contexts outside China, which could affect its results as mentioned before; polymorphisms of VEGF affect the effectiveness of bevacizumab which cannot be ruled out for this drug. Another explanation concerning the lack of clinical studies with endostatin could be the uncertainty of its mechanism of action, so its interactions with other drugs remain uncertain.

Furthermore, as previously reported in this review, Mohajeri et al. [39] reported the Challenges of Recombinant Endostatin in Clinical Application especially, highlighting problems in gaining endostatin stability and finding the correct drug delivery carrier. Strategies as the PEGylation of this molecule have been tested, with the aim of increasing its half-life and stability. In vitro studies have shown similar effectiveness as recombinant endostatin, but with a higher half-life [97] and showed no serious side effects on mice [98]. Additionally, an approach with liposome encapsulation of endostatin has shown favorable results in increasing its half-life [99]. Regarding the costs in the production of this molecule, they could decrease as advances in techniques and industry will make it cheaper to produce.

Despite the problems cited above, endostatin has shown potential as an alternative therapy to anti-VEGF monoclonal antibodies for its effectiveness in advanced stages of cancer, mainly in the context of NSCLC, showing similar efficacy and a good safety profile. Indeed, targeting angiogenesis with monoclonal antibodies has been a great advancement in cancer therapy, however, it still has an important associated toxicity that continues to be a problem, forcing some patients to leave therapy.

Therefore, we believe that endostatin can be a precious asset in the oncological field, both as an adjuvant therapy and as a monotherapy. Thus, we believe that the major guidelines to follow for endostatin to be a drug considered for use are to demonstrate its effectiveness in another setting outside of China and to continue to search for a viable option to increase the half-life of it. Indeed, this molecule could be an option in patients whose clinical characteristics do not favor the use of anti-VEGF antibodies, or whose cancer is resistant to them.

## Figures and Tables

**Figure 1 biomedicines-11-00718-f001:**
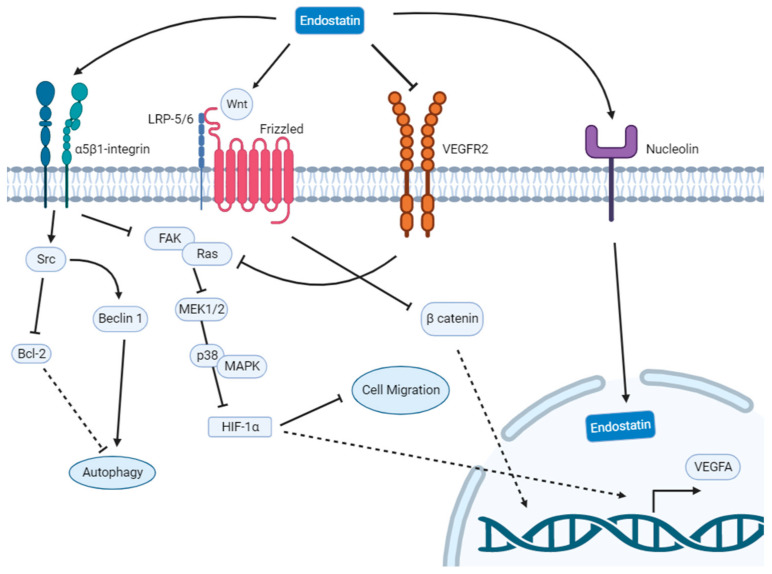
Scheme summarizing the main actions of endostatin on endothelial cells. Bcl-2, B-cell leukemia/lymphoma 2 protein; FAK, Focal adhesion kinase; HIF-1α, Hypoxia induced factor 1-alpha; LRP-5/6, Leucine-responsive regulatory protein; MAPK, Mitogen-activated protein kinase; MEK1/2, Mitogen activated protein kinase kinase; VEGFA, Vascular endothelial growth factor A; VEGFR2, Vascular endothelial growth factor receptor 2.

**Table 1 biomedicines-11-00718-t001:** A summary of relevant articles testing endostatin.

Study Design	Radio/Chemotherapy Used	Endostatin Scheme	Main Outcomes	Adverse Events	Ref.
Breast Cancer					
Phase 2 RCT, *n* = 67, stage IIA to IIIC, first-line treatment	DTX EPR	7.5 mg/m^2^ for 14 days every 3 weeks	ORRs 90.9% vs. 67.7% in control group pCR was identified in 5 (15.2%) vs. 2 (6.5%)	No significant difference was found between groups.	[89]
Phase 2 RCT, *n* = 87, stage III,	DTX EPR CTX	15 mg/day i.v. for 14 days every 3 weeks	ORRs 81.82% vs. 58.14% in control group Median RFS 67.3 m vs. 55.0 m in control group Median OS 74.2 m vs. 59.1 m in control group 3- and 5-year OS 88.5% and 82.8% vs. 76.7% and 54.4% in control group	No significant difference was found between groups.	[84]
Phase 3 RCT, *n* = 803, stage IIA to IIIC BC, first-line treatment	DTX EPR	7.5 mg/m^2^ for 14 days every 3 weeks	ORRs 91.0% vs. 77.9% in control group pCR was identified in 43 (10.7%) patients vs. 31(7.7%) in the control group, but no significant difference was found.	No significant difference was found between groups.	[88]
RCT, *n* = 64, stage IIA to IIIC	DTX EPR	7.5 mg/m^2^ i.v. for 14 days every 3 weeks	ORRs 90.9% vs. 67.7% in control group Mean tumor size change was 21.18 cm^3^ ± 7.32 vs. 15.95 cm^3^ ± 4.32 in control group	Not reported	[90]
Prospective study, *n* = 57, HER-2 negative metastatic BC	TAX ABX TAX + GEM DTX + CPC	7.5 mg/m^2^ i.v. for 14 days and was continued until progressive disease	ORRs was 68.4% for the whole population and 79.3%, 54.5%, and 16.7% for first-line, second-line, and third-line or beyond treatment group, respectively. Median PFS was 10.8 m	Grade 3–4 adverse events: neutropenia (80.7%), leukopenia (77.2%), liver dysfunction (10.5%), and peripheral neurotoxicity (8.8%).	[86]
Prospective study, *n* = 21, advanced TNBC	CB/CP + GEM DTX PEM	30 mg/d i.v. for 7 days in addition to the chemotherapy cycle	ORRs was 47.6% for the whole population, while it was 50% and 44.4% for first-line and second-line or beyond treatment group, respectively. OS 13.3 months PFS 8.8 months	Grade 3–4 adverse events: neutropenia (14.3%), anemia (14.3%), leukopenia (9.5%), thrombocytopenia (9.5%), febrile neutropenia (4.8%), and hypertension (4.8%).	[87]
Colorectal cancer					
Pilot study, *n* = 24, metastatic CCR and GC	CAPIRI GP XELOX DCF FOLFIRI FOLFOX4	15 mg daily i.v. for 14 days every 3 weeks or 7 days every 2 weeks	ORR 19% and 57.1% in first-line treatment group Disease control rate 47.6%	Grade 3–4 adverse events: Leukopenia (30.4%), Neutropenia (34.8%), Thrombocytopenia (17.4%), Anemia (13.0%) Cardiac adverse events: Three patients presented transient sinus bradycardia with spontaneous remission.	[74]
Retrospective controlled study, *n* = 36, no metastatic CCR	FOLFOX-4	15 mg daily i.v. for 14 days	ORR 38.9% vs. 22.2% in control group OS 12.1 m vs. 11.4 m in control group PFS 6.4 m vs. 3.8 m in control group	Grade 3–4 adverse events: Leukopenia (16.7% vs. 11%) Thrombocytopenia (5.6% vs. 0%) Nausea/vomiting (5.6% vs. 5.6%) Cardiac adverse events Hypertension 17.7% vs. 5.6% in the control group Cardiac ischemia 11.1% vs. 0.0% in the control group.	[71]
RCT, *n* = 38, advanced CRC	IR + 5FU + CF	15 mg i.v. daily	OR 42.9% vs. 29.4% in control group median TTP 14.5 months vs. 11 in control group	Cardiac adverse events: Three patients presented grade 1 electrocardiogram abnormalities and two presented grade 1–2 hypertension. No significant difference in the other adverse events between groups.	[73]
Phase 1 trial, *n* = 21, advanced CRC	OX + FU + FA (FOLFOX)	ascending doses from 7.5 to 75 mg/m^2^/day	Endostar was generally safe and well tolerated	Grade 3–4 adverse events: Neutropenia (23.8%), leucopenia (9.5%), thrombocytopenia (4.8%). Cardiac adverse events: Two patients presented ventricular arrhythmia.	[72]
Upper gastrointestinal tract cancer					
RCT, *n* = 96, gastric cancer with liver metastases	FU + OX + CF	15 mg i.v. in addition to chemotherapy cycles	ORR 70.8% vs. 47.9% in control group OS and PFS significantly higher in E supplemented group	No significant difference in adverse events between groups.	[91]
RCT, *n* = 38, metastatic squamous cell carcinoma of the esophagus	5FU + CP + RT	15 mg i.v. daily	1-year OS 72% vs. 50% in control group Median survival 18.2 m vs. 11.6 m in control group	No significant difference in adverse events between groups.	[67]
Lung cancer					
Phase 2 trial, n= 126, treatment-naive NSCLC	TAX + CPT	7.5 mg/m^2^/d i.v.	ORR 39.3% vs. 23% in control group DCR: 90.2% vs. 67.2% in control group	Slight decrease in overall incidence rate of adverse events in E treatment group. Not statistically significant.	[41]
Prospective study, *n* = 50, hypoxia positive stage I-III NSCLC	RT	15 mg i.v.	Total effective rate: 80% vs. 44% in control group	No significant difference in adverse events between groups.	[42]
RCT, *n* = 75, IB-IIIA NSCLC	CP/DXT/CPT/GEM/PEM/NVB following NCCN guidelines, following surgery	7.5 mg/m^2^ i.v.	Average PFS increased by 9.8 m 5-year OS 59.3 vs. 42.1 in control group	No significant difference in adverse events between groups.	[44]
Phase 2 trial, *n* = 50, stage III NSCLC	DXT + CP, followed by RT	7.5 mg/m^2^/d i.v.	PFS 9.9 m 3-year control rate 51% Median OS 24 months	All toxicities were tolerable with proper treatment.	[45]
RCT, *n* = 30, stage IIIA NSCLC	CP + NVB	7.5 mg/m^2^ i.v.	Tumor regression rate increased approximately in 12% vs. control group OS 19 m vs. 16 m in control group	No significant difference in adverse events between groups. No serious adverse events or death were reported.	[46]
Retrospective study, *n* = 71, stage III/IV NSCLC	PEM + GEM + TAX + INN, followed by 2 cycles of RT	15 mg i.v. in addition to chemotherapy cycles	PFS 12 m vs. 7 m in control group Non-specified higher OS vs. control group	CT + E group’s higher OS equaled higher rate of anemia, thrombocytopenia, nausea/vomiting, diarrhea, and fatigue.	[47]
Phase 2 trial, *n* = 73, unresectable NSCLC	EP + CP + RT	7.5 mg/m^2^/24 h 120 h, 14 days/cycle	PFS 13.3 m OS 34.7 m 51 patients achieved objective response	The most common adverse event was leukopenia. 33 patients had grade 3 or more hematologic events.	[92]
Phase 2 trial, *n* = 193, treatment-naive locally advanced NSCLC	EP + CP, DXT + CP or CPT TAX + CP or CPT NVB + CP PEM + CP GEM + CP; RT	7.5 mg/m^2^	mean OS 29.7 m vs. 21.3 m in control group Hazard Ratio between E and control group: 0.697	The incidence of grade 1 and 2 injury was 33.7% and 14.4%; 9.1% and 3.8% vs. 14.4%; in the E supplemented vs. control group, respectively.	[48]
Phase 2 trial, *n* = 69, stage IIIB/IV NSCLC	GEM + CP	7.5 mg/m^2^ i.v.	PFS 6.8 m vs. 4.3 m in control group Survival rate at 12 m 51.6% vs. 38.7% in control group OS 12.4 months vs. 9.8 in control group	The addition of E to GEM/CP did not increase hematological toxicities.	[49]
Retrospective study, *n* = 136, stage IIIB-IV NSCLC	CP + TAX or PEM or GEM or DXT	15 mg/day	ORR 48.5% vs. 29.5% in control group DCR 91.2% vs. 75% in control group	No significant difference in adverse events between groups.	[50]
Retrospective study, *n* = 115, stage IIIB-IV NSCLC	PEM + CP or INN GEM + CP or INN IR + CP	7.5 mg/m^2^	PFS 8.9 m in extended E group vs. 2 m in non-extended group of patients with squamous cell carcinoma OS 27.2 m in extended E group vs. 10.8 m in non-extended group with squamous cell carcinoma	No difference was found between both groups in the incidence of adverse effects. 12.5% and 12.2% had grade 3 or 4 adverse events.	[51]
RCT, *n* = 200, stage IIIB-IV NSCLC	PEM + CP GEM + CP IR + CP PEM + INN INN + GEM	7.5 mg/m^2^ daily	PFS 8 m in extended E group vs. 5.8 m in non-extended group OS 23 m in extended E group vs. 14 m in non-extended group	There were no statistically significant differences in grade 3 to 4 toxicities overall between the 2 treatment groups.	[43]
RCT, *n* = 128, lung adenocarcinoma with malignant pleural effusion	CP + PEM	45 mg intracavitary	Higher and stronger effect on control of the malignant pleural effusion and disease control rate vs. control group	There was no increase of adverse reactions relative to the control group.	[52]
RCT, *n* = 80, NSCLC with brain metastases	RT	7.5 mg/m^2^/day during radiotherapy	Decrease in brain edema on E treated group No significant differences on OS	No significant difference in adverse events between groups. The most common reaction was granulopenia.	[53]
RCT, *n* = 43, NSCLC with brain metastases	RT	30 mg/day	median PFS 8.1 m vs. 4.9 m in control group OS 14.2 m vs. 6.4 m in control group	No significant difference in adverse events between groups.	[54]
Phase 2 trial, *n* = 33, chemotherapy-naive extended SCLC	CP + EP	15 mg i.v.	Median PFS 5 m Median OS 11.5 m	57.6% developed neutropenia.	[55]
Phase 2 trial, *n* = 22, SCLC	EP + CP or CPT	30 mg/day 3 days prior CT and 4 days post CT	PFS 8 m OS 13.6 m ORR 61.9% DCR 95.2%	The main adverse reactions were myelosuppression, albuminuria, nausea, and vomiting. All patients tolerated the treatment.	[56]
Phase 2 trial, *n* = 140, advanced treatment-naive SCLC	EP + CPT	7.5 mg/m^2^ i.v.	PFS 7.3 m vs. 3.9 m in control group ORR 21% higher vs. control group OS similar to control group QOL higher vs. control group	No differences in toxicity vs. control group.	[57]
Head and neck cancer					
Clinical report, *n* = 22, recurrent grade III-IVB NPC	DXT + CP or DXT + CP + 5FU or GEM + CP; + IMRT	105 mg/m^2^	CR was achieved in 20 patients 1-year OS 93.3% 1-year PFS 92.3% 1-year Distant metastasis free-survival 90%	There were no reports of grade 5 toxicities. Incidence of radiation injury was substantially lower than previous studies.	[76]
Phase 2 trial, *n* = 114, locally advanced NPC	DXT + CP followed by CP + IMRT	7.5 mg/m^2^	CR was achieved in 91.1% of patients E trial group improved the complete remission rate of cervical lymph node metastasis	No significant difference in adverse events between groups. The most frequently observed acute toxicities were neutropenia, vomiting, and mucositis	[77]
Retrospective study, *n* = 23, locally advanced NPC	CP + IMRT	15 mg/day	No significant differences in OS, PFS nor ORR	Incidence of xerostomia, difficulty in mouth opening and subcutaneous soft tissue fibrosis was lower on E group.	[79]
RCT, *n* = 44, recurrent metastatic cervical cancer	GEM + CP or DTX+ CP	7.5 mg/m^2^	median PFS 7.2 m vs. 5.1 m in control group	No significant difference in adverse events between groups.	[93]

BC, Breast cancer; CB, Carboplatin; CF, Calcium Folinate; CP, Cisplatin; CPT, Carboplatin; CPC, Capecitabine; CR, Complete Remission; CT, Chemotherapy; CRC, Colorectal cancer; CTX, Cyclophosphamide; DCR, Disease control rate; DTX, Docetaxel; E, Endostatin; EP, Etoposide; EPR, Epirrubicin; FA, Folinic Acid; FU, Fluorouracil; GEM, Gemcitabine; IMRT, Intensity-modulated radiotherapy; INN, Nedaplatin; IR, Irinotecan; m, Months; ORR, Objective response rate; OS, Overall Survival; OX, Oxaliplatin; PEM, Pemetrexed; PFS, Progression-free survival; QOL, Quality of life; NPC, Nasopharyngeal Carcinoma; NSCLC, Non-small cell lung cancer; NVB, Vinorelbine; RCT, Randomized controlled trial; RFS, Relapse free survival; RT, Radiotherapy; SCLC, Small-cell lung cancer; TAX, Paclitaxel; TNBC, Triple negative breast cancer; TTP, Time to progression.

## Data Availability

Not applicable.

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
