# Peer review of "Endostatin and Cancer Therapy: A Novel Potential Alternative to Anti-VEGF Monoclonal Antibodies"

_biomedicines, 2023, doi:10.3390/biomedicines11030718_

Round 1

Reviewer 1 Report

biomedicines-2223966, Endostatin and cancer therapy: A novel potential alternative to anti-VEGF monoclonal antibodies

The manuscript presents an interesting review on endostatin as potential cancer therapy> it is well written, but it could use some improvements.

One major drawback of the manuscript is that the authors are presenting only the problems of Bevacizumab, and generally they are ignoring all the problems of endostatin. This substance is known for almost 30 years, but still is not used. There are many challenges for the clinical use of this molecule. See for example the section “Clinical application challenges” from the article “The Challenges of Recombinant Endostatin in Clinical Application: Focus on the Different Expression Systems and Molecular Bioengineering” Adv Pharm Bull. 2017 Apr; 7(1): 21–34. Also, the mechanism of action for endostatin is not very well known and many specialist consider this a problem. The mention article is just one of some others. Please check the literature and present all the issues and problems, not only the information that is favorable to you.

A good review should provide insights beyond a simple summary or a collection of data. It feels that the authors are not critical with their data and don’t provide a real analysis of the information presented.

I would expect some personal opinion of the authors. Do they think this molecule could really be a clinical solution? What are the major direction for the future research that should cover the gaps?

Author Response

We are thankful for your comments, article recommendation and suggestions. It has been clarified that endostatin’s mechanism of action is not well known. We have added to the discussion a portion with endostatin negatives and our personal opinion. A sentence was added to clarify that the mechanism of action of endostatin is not fully understood. If you feel that your comments have not been satisfactorily addressed, we are most receptive to further feedback.

Reviewer 2 Report

The review by Méndez-Valdés G et al. presents a very-well written overview of the available evidence for using endostatin as an alternative to anti-angiogenetic agents. Interestingly, endostatin is an endogenous angiogenesis inhibitor, and as a result, it will potentially be less toxic. However, the authors have not studied another problematic aspect of anti-VEGF agents. These agents present very high variability in clinical response. This variability is known to be mainly driven by genetic (PMIDs: 31752122, 21791631, 18824714, 21204912) and/or exposure (PMID: 32272489) factors. Therefore an endogenous molecule could potentially overcome these problems. I would recommend that authors briefly comment on this topic too.

Author Response

(The authors gave the same response as above.)

Reviewer 3 Report

In this review, the authors discussed endostatin, an endogenous protein that strongly inhibits VEGF expression and angiogenesis against cancer. The manuscript idea is interesting and covers the anticancer studies of Endostatin and cancer therapy. 

I suggest highlighting the effective downstream signaling pathway for anti-VEGF as the molecular target through some diagrams/figures utilizing molecular modeling (docking), especially with the surface receptors. 

Author Response

We are grateful for your comments and suggestions, but regarding the latter, I am afraid to report that we could not solve it successfully as we have no expertise in the subject of docking, we tried to search for endostatin but had no luck, and your review came to us after the others. If you believe that it would be of great benefit to us to add what was suggested to the review, it would be of immense help if you could suggest where to search and we will certainly add it.

Round 2

Reviewer 1 Report

The authors improve their paper and corrected the problems presented in the first review. There are some editing problems that need the attention of the authors.